SOFTWARE

# Expanding and improving analyses of nucleotide recoding RNA-seq experiments with the EZbakR suite

Isaac W. Vock[1,2]*, Justin W. Mabin[3], Martin Machyna[1,2¤], Alexandra Zhang[1,2], J. Robert Hogg[3], Matthew D. Simon[1,2]

**1** Department of Molecular Biophysics and Biochemistry, Yale University, New Haven, Connecticut, United States of America, **2** Institute of Biomolecular Design and Discovery, Yale University, West Haven, Connecticut, United States of America, **3** Biochemistry and Biophysics Center, National Heart, Lung, and Blood Institute, National Institutes of Health, Bethesda, Maryland, United States of America

¤ Current address: Paul-Ehrlich-Institut, Host-Pathogen-Interactions, Langen, Germany
* isaac.vock@yale.edu

## Abstract

Nucleotide recoding RNA sequencing methods (NR-seq; TimeLapse-seq, SLAM-seq, TUC-seq, etc.) are powerful approaches for assaying transcript population dynamics. In addition, these methods have been extended to probe a host of regulated steps in the RNA life cycle. Current bioinformatic tools significantly constrain analyses of NR-seq data. To address this limitation, we developed EZbakR (https://github.com/isaacvock/EZbakR), an R package to facilitate a more comprehensive set of NR-seq analyses, and fastq2EZbakR (https://github.com/isaacvock/fastq2EZbakR), a Snakemake pipeline for flexible preprocessing of NR-seq datasets, collectively referred to as the EZbakR suite. Together, these tools generalize many aspects of the NR-seq analysis workflow. The fastq2EZbakR pipeline can assign reads to a diverse set of genomic features (e.g., genes, exons, splice junctions), and EZbakR can perform analyses on any combination of these features. EZbakR extends standard NR-seq mutational modeling to support multi-label analyses (e.g., s⁴U and s⁶G dual labeling), and implements an improved hierarchical model to better account for transcript-to-transcript variance in metabolic label incorporation. EZbakR also generalizes dynamical systems modeling of NR-seq data to support analyses of premature mRNA processing and flow between subcellular compartments. Finally, EZbakR implements flexible and well-powered comparative analyses of all estimated parameters via design matrix-specified generalized linear modeling. The EZbakR suite will thus allow researchers to make full, effective use of NR-seq data.

**Data availability statement:** All code required to reproduce figures and supplemental figures is available on a GitHub repository at https://github.com/isaacvock/EZbakRsuite_paper_code. Data necessary to run this code is available on Zenodo at https://zenodo.org/records/13929898.

**Funding:** This work was supported by funding from the National Institutes of Health grants R01GM137117 (NIH; https://www.nih.gov/), awarded to MDS, and T32GM67543-19, awarded to IWV. It was also funded by the National Heart, Lung, and Blood Institute grant ZIAAHL006158 (NHLBI; https://www.nhlbi.nih.gov/) awarded to JRH. This work was also supported by the Intramural Research Program, National Heart, Lung, and Blood Institute, National Institutes of Health, and utilized the computational resources of the NIH HPC Biowulf cluster (http://hpc.nih.gov). The funders did not play any role in the study design, data collection and analysis, decision to publish, or preparation of the manuscript.

**Competing interests:** The authors have declared that no competing interests exist.

## Author summary

Metabolic labeling is a powerful tool for tracking the rate at which RNAs are made, processed, and degraded. It involves treating cells with a modified nucleotide that is added as a building block into RNAs during the cell's exposure to the nucleotide. While the modified building block is largely invisible to the cell, the identity of the labeled RNA molecules can be revealed using chemistry that recodes the hydrogen bonds of the label, thereby introducing apparent mutations in sequencing reads from labeled RNA. These nucleotide recoding/conversion RNA-seq (NR-seq) experiments require software tools to process and analyze NR-seq data, but existing tools lack the flexibility needed to best use these data. To address this gap, we developed the EZbakR suite, comprising a data processing pipeline (fastq2EZbakR) and an R package (EZbakR). The EZbakR suite helps users analyze their data in many ways (e.g., at the level of genes or isoforms), improves the quantification of labeled RNA, and supports complex experimental designs (e.g., subcellular fractionation, multi-factor perturbations). Its modularity also affords the flexibility to further extend and improve NR-seq analyses. We thus anticipate that the EZbakR suite will allow analyses of a wide range of NR-seq experiments to uncover new RNA biology.

## Introduction

Developing a mechanistic understanding of gene expression regulation requires methods to probe the kinetics of RNA synthesis, processing, transport, and degradation. Standard RNA-seq provides limited information about the kinetics of the processes that determine the abundance of an RNA. Nucleotide recoding RNA-seq (NR-seq; a.k.a. nucleotide conversion RNA-seq; TimeLapse-seq, SLAM-seq, TUC-seq, etc.) overcomes these limitations by combining metabolic labeling with chemistries that recode the hydrogen bonding pattern of the metabolic label to facilitate detection of labeled RNA via chemically induced mutations in sequencing reads [1–3]. By providing information about both overall RNA abundance and the dynamics of nascent and pre-existing RNA, NR-seq resolves the kinetic ambiguities of standard RNA-seq [4,5].

The field has seen an explosion of methodologies using NR-seq to study different aspects of the RNA life cycle. Nucleotide recoding has provided unique kinetic insights via integration with methods such as Start-seq [6,7], Ribo-seq [8,9], PacBio long read sequencing [10,11], subcellular fractionation [12–14], and single cell RNA-seq [15–20]. In addition, NR-seq has applications beyond quantifying RNA kinetics, and has been used to probe small molecule binding sites across the transcriptome [21], filter out reads from unlabeled RNA contamination in enrichment-based strategies [3,22], and improve analyses of single cell trajectory inference [20,23,24]. Finally, while nucleotide recoding chemistries were originally developed for 4-thiouridine (s⁴U), some have been shown to successfully recode 6-thioguanosine (s⁶G) as well

[25,26]. This has opened the door for multi-label experimental designs [27]. This family of NR-seq extensions has already provided a host of novel biological insights, and its continued growth creates a need for better and more accessible tools for NR-seq data to expand our understanding of RNA biology.

We aimed to develop a suite of tools that can:

1. Analyze reads assigned to a flexible array of genomic features (e.g., exons, introns, exonic bins, splice junctions, transcript equivalence classes). This would facilitate integration with differential expression analyses capable of working with the same feature sets and allow users to make full use of all their aligned NR-seq reads.

2. Make the processed mutational data readily available to users to support assessment of model fits and future model improvements.

3. Implement a rigorous statistical model of the mutational data in NR-seq reads to estimate the abundances of labeled and unlabeled RNA. These models should also be compatible with multi-label data.

4. Support flexible kinetic modeling that is compatible with standard analyses of mature total RNA as well as analyses of premature mRNA dynamics and flow between subcellular compartments to better support the full range of NR-seq methods.

5. Perform well-powered and flexible comparative analyses of any estimated kinetic parameter to enable in-depth investigations of experimental perturbations.

To date, no existing tool has all these capabilities. pulseR, originally designed for enrichment-based methods, has been repurposed for NR-seq data, but may not be suitable for enrichment-free data, and lacks data processing functionality [28,29]. SLAMDUNK, one of the first pipelines developed to process raw NR-seq data, is widely used but was specifically optimized for single-end, 3′-end sequencing data [1,30]. In addition, it uses a mutation count cutoff to distinguish labeled and unlabeled reads, which can yield biased estimates of labeled RNA abundance, depending on the incorporation levels of $s^4U$ versus U in labeled RNAs in a particular sample (Fig A in S1 Text). A gold-standard analysis pipeline must overcome the limitations of this labeled vs. unlabeled read classification approach.

Our lab developed a two-component mixture model to estimate the fraction of sequencing reads derived from labeled RNA [3]. This strategy was subsequently implemented in the user-friendly and performant software tool GRAND-SLAM [31]. GRAND-SLAM performs analyses at the gene-level, allowing users to either retain or throw out reads mapping to intronic regions. Separately analyzing premature and mature mRNA dynamics with GRAND-SLAM is challenging and requires custom workarounds [12]. In addition, GRAND-SLAM provides mixture model fits as output, but provides limited access to the intermediate mutational data passed to the mixture model. Providing such data in a convenient format would help users explore their mutational data and facilitate development of improved modeling strategies [12–14,32]. While the recently developed R package grandR greatly facilitates many downstream analyses of NR-seq data, it is not designed to overcome these key limitations [33].

We previously addressed some of these shortcomings with the development of a pipeline (bam2bakR) and R package (bakR) [34]. The output of bam2bakR includes a compressed representation of the processed mutational data, which can be analyzed via mixture models implemented in bakR. In addition, bakR borrows strategies from differential expression analysis tools to perform well-powered comparative analyses of kinetic parameters and better support downstream biological investigations [35,36]. While bam2bakR can separately assign reads to exonic and intronic regions, bakR provides limited support for making use of that information to study pre-mRNA dynamics, as it can only perform analyses on one feature set at a time (e.g., exonic regions of genes). bakR's rigid design and lack of modularity also limits the type of NR-seq datasets it can analyze. Finally, no existing tool supports analyses of multi-label experiments or flexible dynamical systems modeling of NR-seq data. The field needs further bioinformatic innovation to make more effective use of NR-seq data.

Here we present the EZbakR suite, a Snakemake pipeline (fastq2EZbakR) and R package (EZbakR) designed to support flexible analyses of NR-seq experiments (Fig 1). fastq2EZbakR assigns reads to a wide array of genomic features, and EZbakR supports analyses of any combination of these features. EZbakR generalizes the mutational mixture model implemented in GRAND-SLAM and bakR to support multi-label designs. It also optionally implements an improved hierarchical model that allows individual features to have unique incorporation rate estimates, addressing a recently discovered source of variance in NR-seq data [32]. In addition, EZbakR generalizes fitting models of linear dynamical systems to NR-seq data to support analyses of pre-mRNA dynamics and of subcellular fractionation-based extensions of standard NR-seq [12–14]. Finally, EZbakR implements design matrix specified comparative analyses of kinetic parameters to radically increase its flexibility relative to its predecessor, bakR. We use a panel of simulated data to validate these novel functionalities and showcase how the EZbakR suite promises to facilitate unprecedented investigations of existing and future NR-seq datasets.

## Design and implementation

### fastq2EZbakR

fastq2EZbakR is a Snakemake pipeline that can accept either fastq files or aligned bam files (the latter making fastq2EZbakR compatible with specialized alignment strategies not directly implemented in our pipeline [37]) as input (Fig 2B) [38]. It produces as output a file we term a counts Binomial (cB) file, which contains information about the mutation content of reads and the features they were assigned to (see Fig 1, highlight 2 for simplified schematic). If fastq files are provided as input, adapters are first trimmed with fastp and reads are aligned with either STAR or HISAT2 (user choice, STAR by default given recent benchmarks showing it performs similarly to specialized 3-base alignment strategies) [39–43]. In all cases, unmapped reads and non-primary alignments are filtered from the resulting or provided bam file with SAMtools [44]. Mutations are counted with a custom Python script, and feature assignment is performed as described in the S1 Text under *Generalized feature assignment in fastq2EZbakR* (all strategies are schematized in Fig 2A), using a combination of featureCounts and custom scripting [45]. The mutation counts and feature assignment tables are merged and combined for all samples to give the final cB file. Documentation detailing how to run fastq2EZbakR can be found here: https://fastq2ezbakr.readthedocs.io/en/latest/

### Generalized mixture modeling in EZbakR

Mutational mixture modeling in EZbakR is performed in two steps:

1. Mutation rates in labeled and unlabeled reads ($p_{labeled}$ and $p_{unlabeled}$) for all types of mutations being modeled are estimated. This is done by fitting a two-component mixture model for each mutation type to all reads from a given sample. Fitting is done for each mutation type independently at this stage. If users have control samples lacking any labeling, they can optionally choose to use a single $p_{unlabeled}$ derived from these samples. This can improve the stability of $p_{labeled}$ estimates in some settings, like when incorporation rates are low.

2. The generalized mixture model parameters are estimated via the method of prior-penalized maximum likelihood, similar to as previously described for bakR [34].

   See S1 Text for further details and mathematical formalization (formalization is also succinctly depicted in Fig 3A).

### Hierarchical mixture model in EZbakR

Hierarchical mixture modeling in EZbakR is currently only compatible with standard two-component mixture modeling (Fig 4A). This strategy starts with the same step 1 as in the generalized mixture modeling described above. After global estimates for $p_{labeled}$ and $p_{unlabeled}$ are obtained, the following iterative strategy is used to perform maximum likelihood estimation with feature-specific $p_{labeled}$ values:

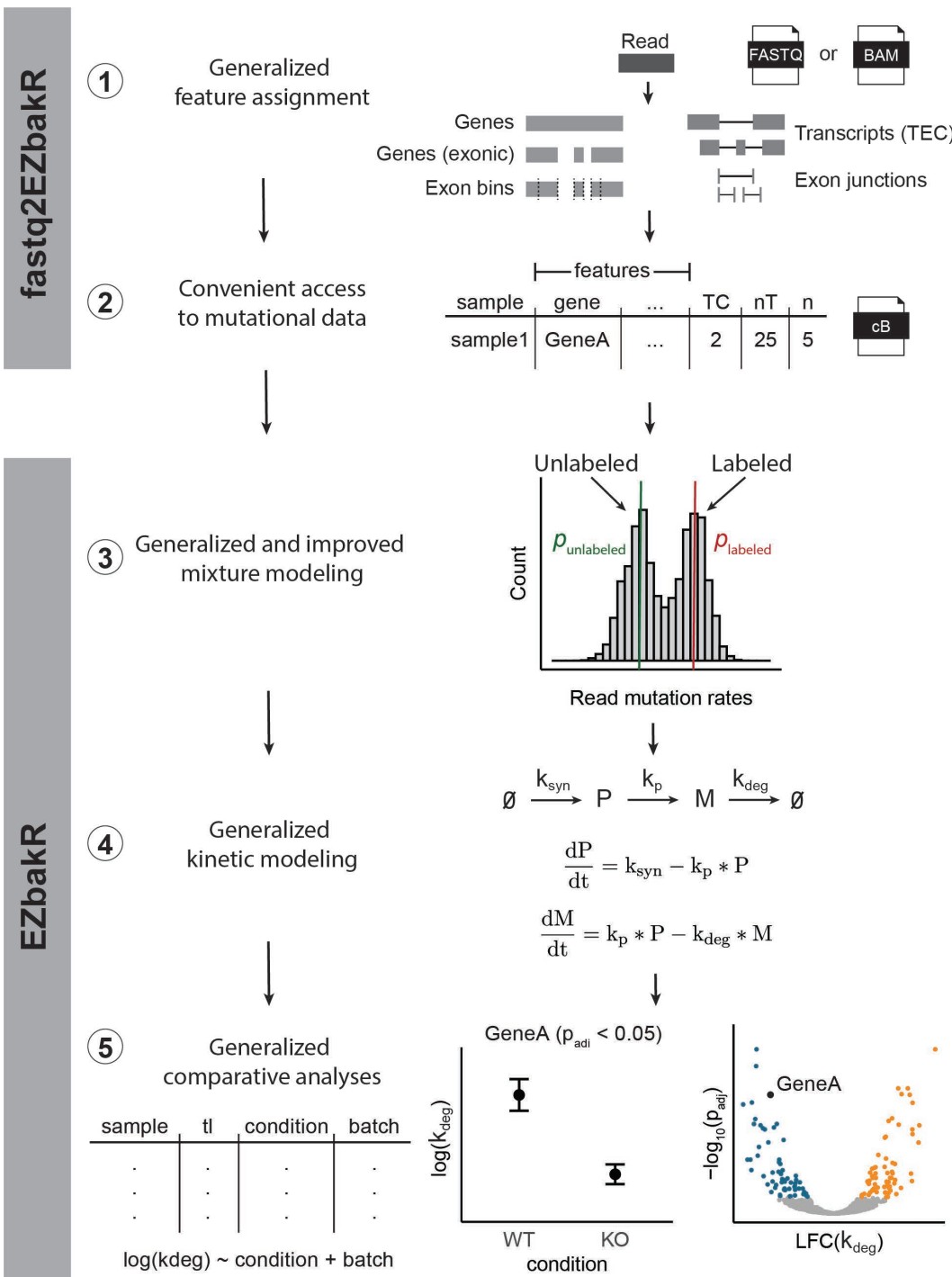

**Fig 1. The EZbakR suite generalizes and improves all steps of the NR-seq analysis pipeline.** The EZbakR suite: 1) Implements a flexible feature assignment strategy, 2) provides processed mutational data in a convenient, compressed format, 3) analyzes mutational data in a way that supports multi-label design and allows for feature-to-feature mutation rate variance, 4) fits any identifiable, linear dynamical systems model to NR-seq data, and 5) performs well-powered, design matrix-specified comparative analyses of all estimated kinetic parameters.

**A)**

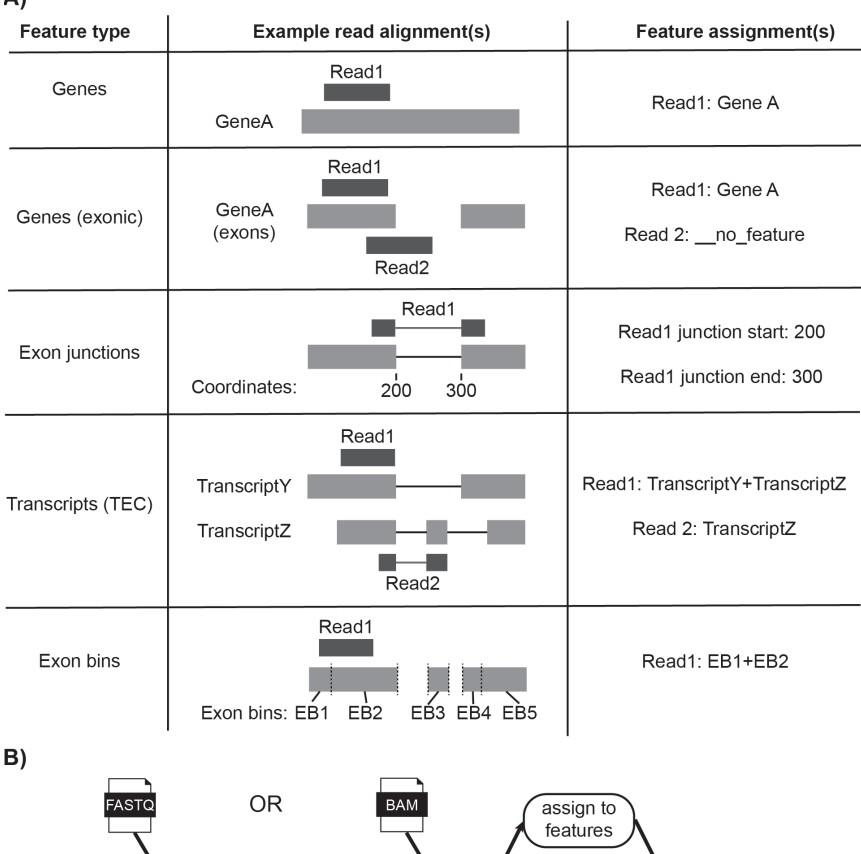

| Feature type | Example read alignment(s) | Feature assignment(s) |
|---|---|---|
| Genes | Read1 / GeneA | Read1: Gene A |
| Genes (exonic) | Read1 / GeneA (exons) / Read2 | Read1: Gene A / Read 2: __no_feature |
| Exon junctions | Read1 / Coordinates: 200 300 | Read1 junction start: 200 / Read1 junction end: 300 |
| Transcripts (TEC) | Read1 / TranscriptY / TranscriptZ / Read2 | Read1: TranscriptY+TranscriptZ / Read 2: TranscriptZ |
| Exon bins | Read1 / Exon bins: EB1 EB2 EB3 EB4 EB5 | Read1: EB1+EB2 |

**B)**

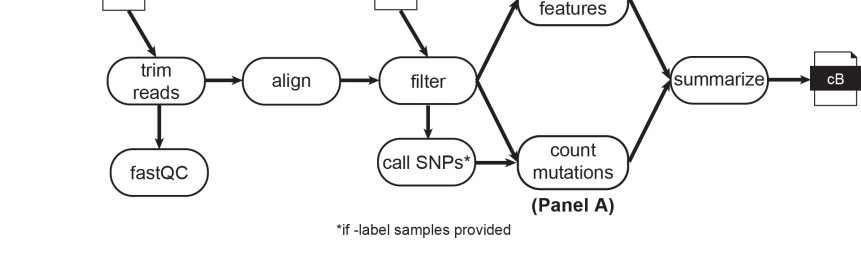

*if -label samples provided

**Fig 2. fastq2EZbakR generalizes the assignment of reads to features to support finer dissection of NR-seq data.** A) Schematic of the 5 different feature assignment strategies implemented in fastq2EZbakR. If a read does not overlap with a given feature, it will be assigned a value of __no_feature for that feature assignment. If a read overlaps multiple features, all features will be reported, with names separated by +-signs. TEC = transcript equivalence class (set of transcript isoforms with which a read is compatible). Exon bins were introduced in DEXSeq (Anders et al., 2014). B) Schematic of the full fastq2EZbakR pipeline.

1. A two-component mixture model with feature-specific $p_{labeled}$'s is fit to all features with more than a certain number of reads (300 by default) in each sample. Low coverage features are not considered in this step as the limited number of reads for such features precludes stable estimation of a feature-specific $p_{labeled}$.

2. A strongly regularizing logit($p_{labeled}$) prior is estimated for each sample from the distribution of feature-specific $p_{labeled}$'s estimated in the previous step. More specifically, a normal distribution prior is inferred with mean equal to the estimated sample-wide logit($p_{labeled}$) and standard deviation equal to the standard deviation of feature-specific logit($p_{labeled}$) estimates minus the average feature-specific logit($p_{labeled}$) uncertainty. If this difference is less than 0, a user-defined small standard deviation (0.01 by default) is imputed. If this difference is greater than a user-defined maximum (0.15 by

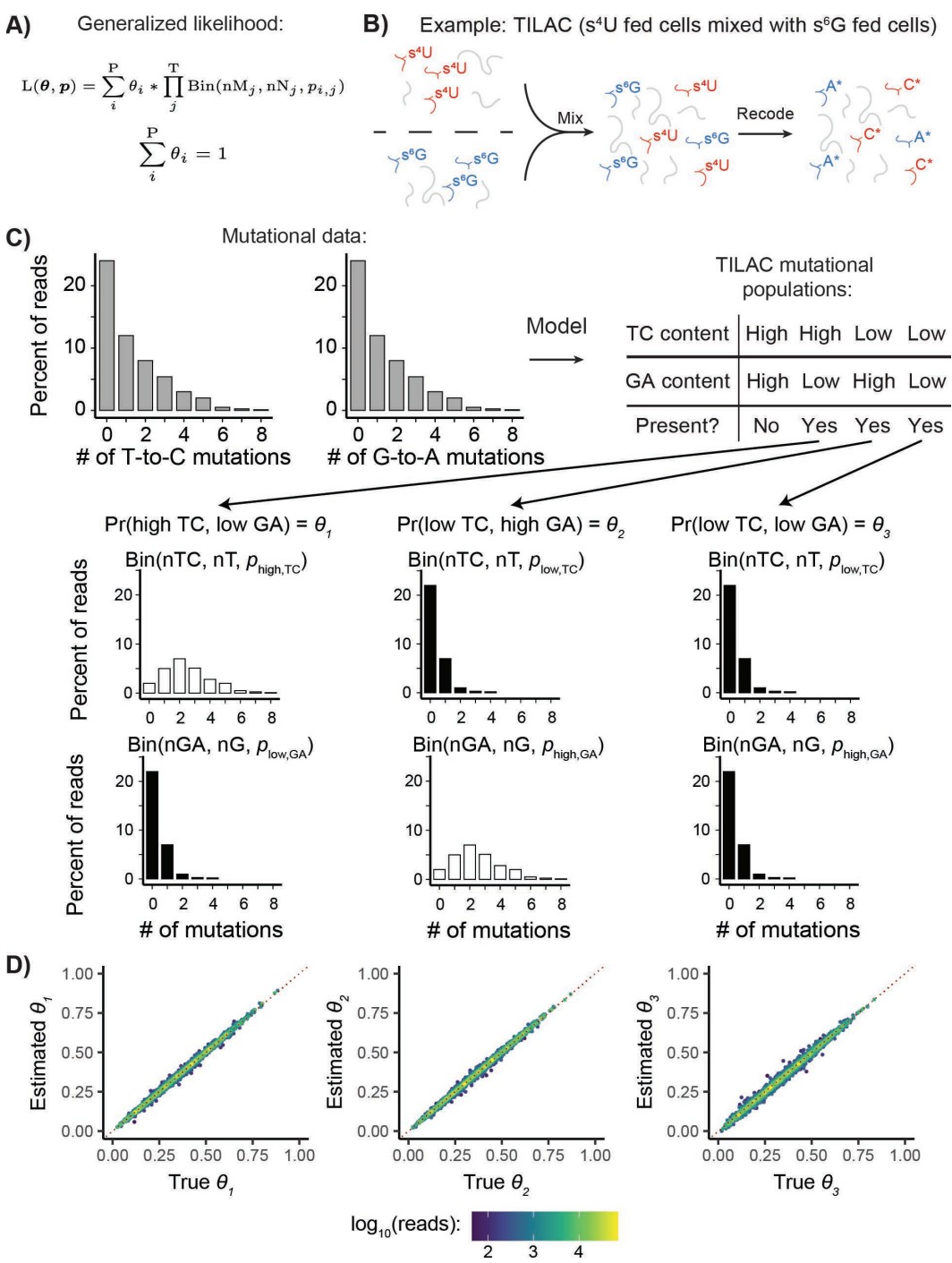

**Fig 3. EZbakR generalizes NR-seq mixture modeling to support multi-label analyses. A)** Generalized mixture model likelihood. P = number of distinct mutational populations (e.g., high T-to-C and low G-to-A mutation rate). T = number of mutation types being analyzed (e.g., T-to-C and G-to-A). nM = number of mutations of a particular type in a given read. nN = number of mutable nucleotides of a given type in a given read. **B)** Example of a dual-label NR-seq experimental method: TILAC. In this experiment, s⁴U fed cells are mixed with s⁶G fed cells. **C)** Schematic for how generalized mixture modeling works in the setting of TILAC. In TILAC, there are no dually labeled reads, so the high T-to-C and high G-to-A population does not exist (see mutational populations table). **D)** Analyses of simulated TILAC data. $\theta_1$ = fraction s⁴U labeled; $\theta_2$ = fraction s⁶G labeled; $\theta_3$ = fraction unlabeled. X-axis is simulated ground truth. Y-axis is estimated value. The red dotted line represents perfectly accurate estimation.

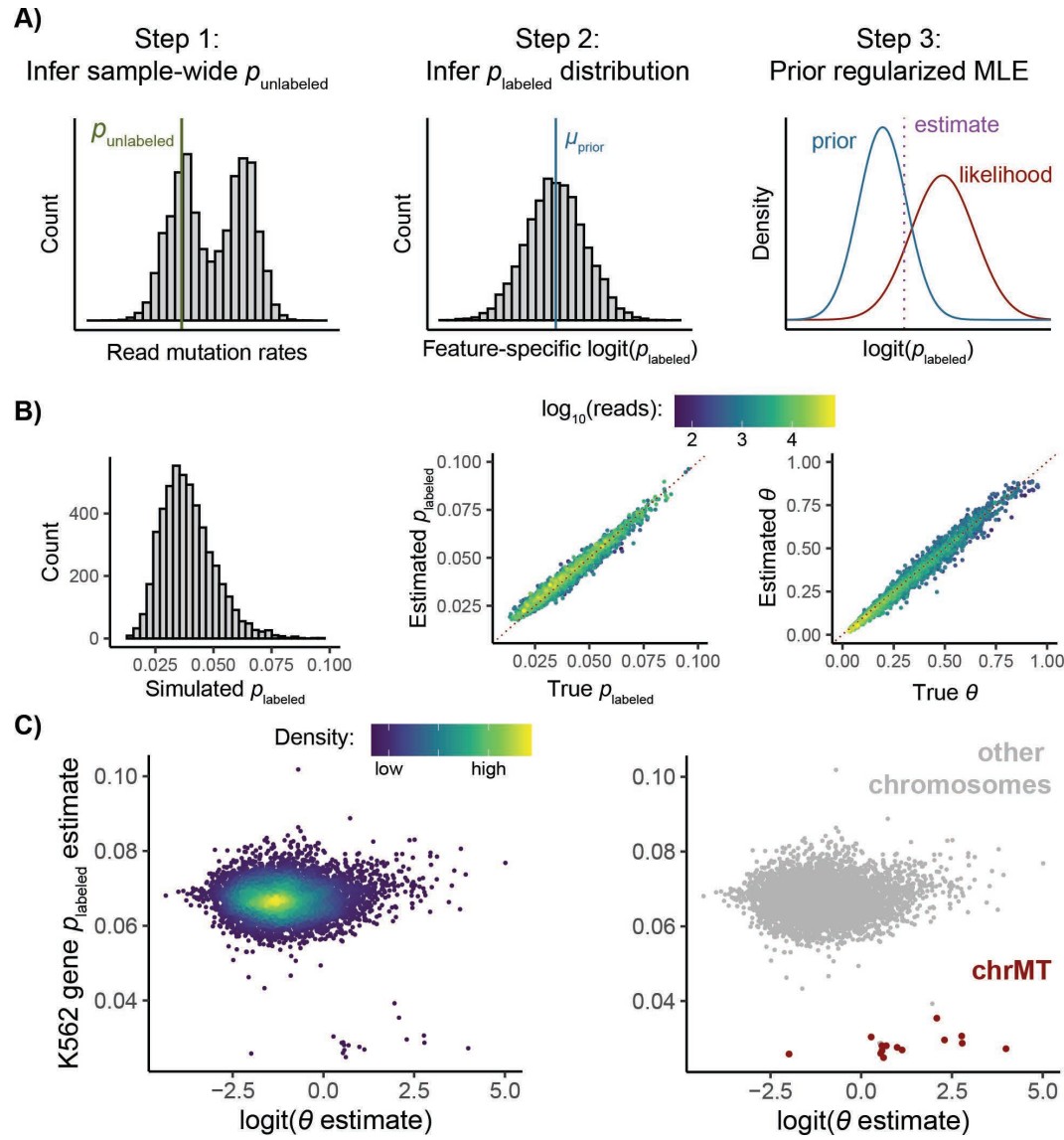

**Fig 4. EZbakR's hierarchical NR-seq mixture modeling accounts for $p_{labeled}$ variation. A)** A schematic of the hierarchical modeling strategy to infer a $p_{labeled}$ for each feature (i.e., feature-specific $p_{labeled}$). Strategy is designed to strongly regularize feature-specific $p_{labeled}$ estimates to reduce estimate variance. See Design and Implementation for details. **B)** Analyses of simulated data. Left: distribution of simulated feature-specific $p_{labeled}$. Middle: assessment of feature-specific $p_{labeled}$ estimate accuracy. Right: Assessment of fraction labeled ($\theta$) estimate accuracy. In Middle and Right plots, the red, dotted line represents perfect estimation. Points are colored by simulated read count. **C)** Estimated gene-specific $p_{labeled}$ (Y-axis) as a function of the estimated fraction labeled (on a logit-scale; X-axis) from analysis of TimeLapse-seq data from K562 cells (Ietswaart et al., 2024) [12]. Left: points colored by density. Right: points colored by whether the gene is on the mitochondrial chromosome (chrMT).

default), the maximum is imputed. Extreme estimates (i.e., those at the set parameter bounds of between -9 and 0 on the logit scale) are filtered out before calculating the mean and standard deviation of the distribution. If multiple labeled samples are being modeled, the minimum sample-specific prior standard deviation is used for all samples so as to avoid under-conservativeness that could yield highly unstable estimates in the next step.

3. A two-component mixture model with feature-specific $p_{labeled}$'s is fit to all features with the strongly regularizing logit($p_{labeled}$) prior obtained in step 2.

## Generalized linear dynamical systems modeling in EZbakR

Generalized linear dynamical systems modeling in EZbakR is performed with the EZDynamics() function. It can take as input either sample-specific fraction labeled estimates (output of EZbakR's EstimateFractions() function) or condition-wide averages obtained from the EZbakR generalized linear model fit (AverageAndRegularize() function). Both are compatible with modeling of pre-mRNA dynamics, but modeling of RNA flow between subcellular compartments is only compatible with the latter. This is because modeling of subcellular compartment flow requires integrating across multiple independent samples (i.e., data from different subcellular fractions), so there is no way to estimate all kinetic parameters of such models for individual samples. The input to EZDynamics() is a matrix representation of graphical models akin to those in Fig 5. Documentation regarding model specification can be found here: https://isaacvock.github.io/EZbakR/articles/EZDynamics. html. Other input includes information about how the measured features relate to the modeled features. For example, if fitting the subcellular fractionation model shown in Fig 5C, you will likely have whole cell data. The RNA from this sample corresponds to the sum of N (nuclear RNA) and C (cytoplasmic RNA), and this fact must be conveyed to EZDynamics(). EZDynamics() then uses the method of maximum likelihood to estimate kinetic parameters of your model, and uncertainties are calculated as the square root of the diagonal elements of the inverse Hessian matrix [46]. This is done by modeling both the fraction labeled and, if rigorous normalization is possible, the read counts for a given feature. See S1 Text for details.

Estimating scale factors for multi-compartment modeling can be done in one of three ways:

1. Spike-in normalization. EZbakR allows users to input spike-in derived scale factors.

2. Fraction labeled modeling. The fraction labeled estimates (which are internally normalized) can be used to estimate all but the RNA synthesis parameter, if such an estimation strategy is identifiable. Scale factors can then be inferred from the set of non-synthesis parameter estimates, by identifying the factors by which absolute RNA abundances are expected to differ across the compartments (since the synthesis rate is the same for all RNA species from a given gene). A downside of this strategy is that it limits the complexity of identifiable models and typically yields lower confidence parameter estimates (Fig D in S1 Text, top row).

3. Fraction labeled mixing model. If users have data for individual subcellular compartments as well as data for a sufficient number of combinations of these compartments, then scale factors can be estimated from the differences in overall fraction of reads that are labeled in each compartment or combination of compartments. For example, if users have nuclear fraction, cytoplasmic fraction, and whole cell data, then normalization factors can be estimated by solving a linear system of equations and using the inferred ratio of absolute molecular abundances to derive scale factors for each compartment. See S1 Text for details and mathematical formalization.

Option 3 was used in analyses of subcellular fractionation data in Figs 5, D, E, and G of S1 Text. Option 2 was used in the subcellular analyses in Fig F of S1 Text due to the more complex model being unnormalizable by method 2 (without directly probing the free nucleoplasmic and cytoplasmic RNA, the mixing model cannot be fit), and the data lacking spike-ins. For the subcellular analysis in Fig I in S1 Text, data provided by Tarrero et al was used, which was normalized via a strategy discussed in that work (thus strategy 1 was used in this case) [47].

## Generalized linear modeling in EZbakR

EZbakR's AverageAndRegularize() function is used to fit a generalized linear model of a user's fraction labeled or kinetic parameter estimates. Parameter standard errors are then regularized using a hierarchical modeling strategy similar to that introduced in bakR (Vock and Simon 2022). See Supplemental Methods in S1 Text for a discussion of the improvements made to this regularization scheme in EZbakR. The EZbakR input data object (an EZbakRData object) includes a so-called metadata data frame (metadf), which can contain any number of factors describing elements of each sample

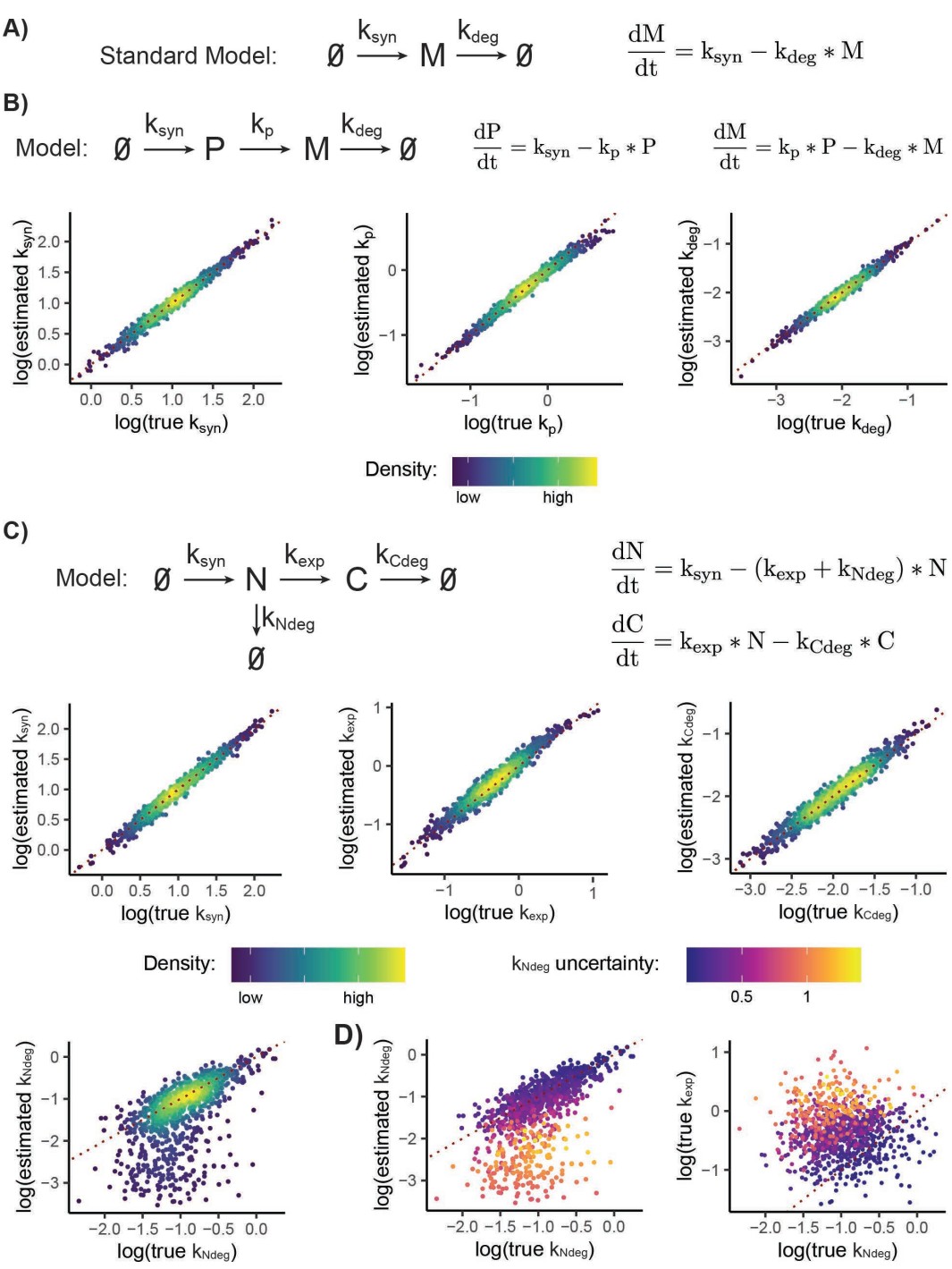

**Fig 5. EZbakR generalizes kinetic modeling of NR-seq data. A)** Model assumed when performing standard analysis of mature mRNA synthesis and degradation. M = mature mRNA. **B)** Analysis of simulated data for model of pre-mRNA maturation. P = premature mRNA; M = mature mRNA. Scatter plots show comparison of true simulated parameter values to those estimated by EZbakR, for all three kinetic parameters in the model. Red dotted line represents perfect estimation. **C)** Analysis of simulated data for a model of nuclear-to-cytoplasmic trafficking of RNA. N = nuclear RNA; C = cytoplasmic RNA. Red dotted line represents perfect estimation. **D)** Left: Nuclear degradation rate constant accuracy scatterplot from **C**, colored by the model's uncertainty in the rate constant estimate. Right: comparison of the true nuclear degradation and export rate constants, colored by the model's uncertainty in the nuclear degradation rate constant. Red dotted line represents equal nuclear degradation and export kinetics. Estimating $k_{Ndeg}$ is expected to get harder the further points are from this line, for reasons discussed in S1 Text.

(see examples in Figs 6A and H S1 Text). These factors can be included in a formula object passed to AverageAndRegularize(), which will specify the linear model to fit to one's data. A unique aspect of generalized linear modeling in EZbakR is that it can account for heteroskedasticity several different ways. One, if a model is such that each parameter is estimated from a non-overlapping set of samples, then standard deviations are estimated for each of these sample sets independently or two, users can specify a set of factors by which to group samples, and standard deviations are estimated in these sample sets and these standard deviations are used to estimate parameter standard errors. See S1 Text for details.

## Generating simulated data

Simulated data used in this study were generated with simulation functions implemented in EZbakR. Multi-label data was simulated with the SimulateMultiLabel() function in EZbakR. Data with feature-to-feature $p_{labeled}$ variation was simulated with the SimulateOneRep() function in EZbakR. Data for all dynamical systems models tested in Fig 4 were simulated with the SimulateDynamics() function in EZbakR. See documentation and Supplemental Methods in S1 Text for details regarding how these functions work and parameters used. Simulated data for analyses in Fig 6 came from the previously published bakR benchmarks [34].

## Analyses of real NR-seq data with the EZbakR suite

Published NR-seq data from Iestwaart et al. were reprocessed with fastq2EZbakR using default configuration settings, only editing the required strandedness parameter (these data are "forward" stranded) [12]. A cB file from the Courvan et al. data was provided in that paper's GEO submission [48]. cB files were passed along with the necessary metadata data frame (metadf) to EZbakR's EZbakRData() function. EstimateFractions() was then run with default settings, except pold_from_nolabel was set to TRUE when -s4U data was present, or with strategy set to "hierarcial" for analyses using the hierarchical mixture model in Figs 4 and B in S1 Text. If -s4U data was present, dropout was corrected using EZbakR's CorrectDropout() function, which implements a previously described strategy [37]. For the analyses in Fig 7, -s4U data was not included for all subcellular fractions. In this case, we normalized for dropout instead, identifying the lowest dropout sample from each subcellular fraction and using it as the reference to which others were normalized. This strategy is implemented in EZbakR's NormalizeForDropout() function. For analyses in Fig 6F, EZbakR's EstimateKinetics() function was run with default settings, and EZbakR's AverageAndRegularize() function was run using either the model `~state:oxy` or `state*oxy`, the latter specifying inference of the interaction effect plotted in Fig H (panel B) in S1 Text. Volcano plots were made from the relevant output of EZbakR's CompareParameters() function. For analyses in Figs C (panel C), F, and G in S1 Text, AverageAndRegularize() was run specifying logit_fraction_highTC's were to be averaged for each compartment (model: `~compartment`). For analyses in Fig 7, AverageAndRegularize() was run specifying logit_fraction_highTC's were to be averaged for each compartment and siRNA treatment (model: `~compartment:siRNA`). For Figs 7 and C (panel C), F, and G in S1 Text, these averages were fit to the relevant dynamical systems model using EZbakR's EZDynamics() function. Volcano plots in Fig 7 were made from the relevant output of EZbakR's CompareParameters() function.

## grandR analysis

For analyses in Fig 6B-E, grandR's FitKineticsSnapshot() and EstimateRegulation() functions were used with default settings to estimate changes in simulated degradation kinetics. To perform this analysis, estimates of the fraction new from EZbakR had to be written to a table formatted like the main output of GRAND-SLAM so that grandR could import the data. This most notably requires estimating parameters for the fraction new beta distribution posteriors. This was done via fitting a beta distribution to EZbakR's fraction new estimates (posterior mean) and uncertainties (posterior standard deviation) using the method of moments.

grandR's EstimateRegulation() function provides a credible interval (CI) of a user-specified width (95% by default) for each estimated degradation rate constant $\log_2$-fold change (L2FC), as well as a region of practical equivalence (ROPE)

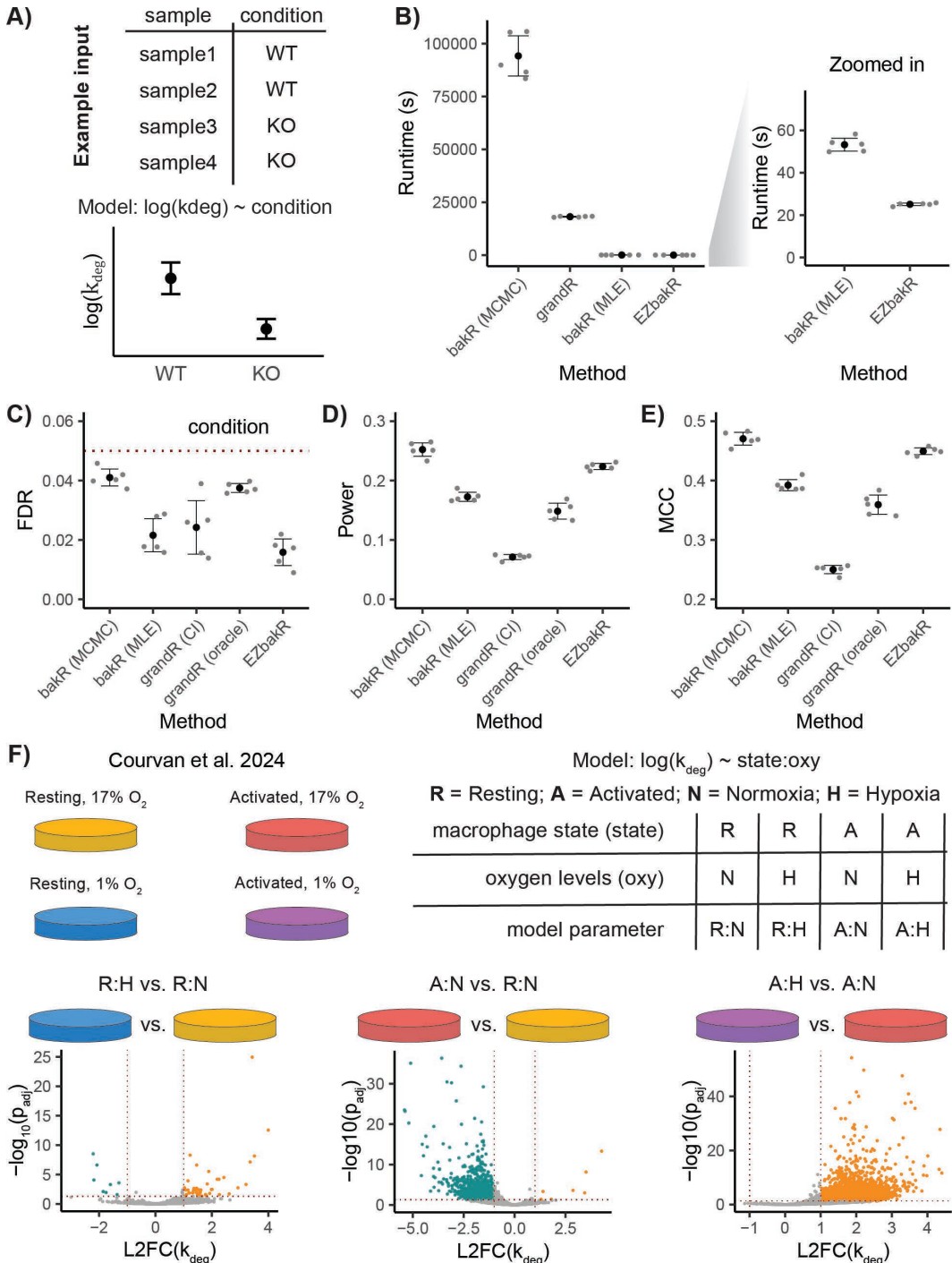

**Fig 6. EZbakR improves and generalizes performing comparative analyses with NR-seq. A)** Simplified input to the generalized kinetic parameter linear model in EZbakR. Includes metadata for each sample analyzed and a model relating a given kinetic parameter to factors included in the metadata. Any identifiable model can be specified and fit. This approach allows for simple multi-condition comparisons (shown here) or more complicated analysis designs (e.g., multi-factor designs; Fig H, panel A, in S1 Text) **B)** Comparison of runtimes between two bakR implementations (Markov Chain Monte Carlo (MCMC) and Maximum Likelihood Estimation (MLE)), grandR, and EZbakR. **C-E)** Analysis of simulated data originally presented in [34]. We include two grandR assessments. In one case (CI) we use grandR's provided 95% credible intervals to determine the significance of degradation rate constant changes. In the other (oracle), we identified a region of practical equivalence (ROPE) probability cutoff that yields FDR control on-par with the highest power method (bakR MCMC). The latter, while infeasible in real data applications, gives grandR the best chance of maximizing its power.

C) Comparison of statistical power (number of true positives/ number simulated positives) between bakR implementations, grandR, and EZbakR. **D)** Comparison of false discovery rates (FDRs; number of false positives/ number of positives) between bakR implementations, grandR, and EZbakR. **E)** Comparison of Matthew's correlation coefficients (MCC) between bakR implementations, grandR, and EZbakR. **F)** Application of the generalized linear model in EZbakR for analysis of a real, multi-perturbation dataset previously analyzed with bakR [48]. Schematics are adapted from those in that study.

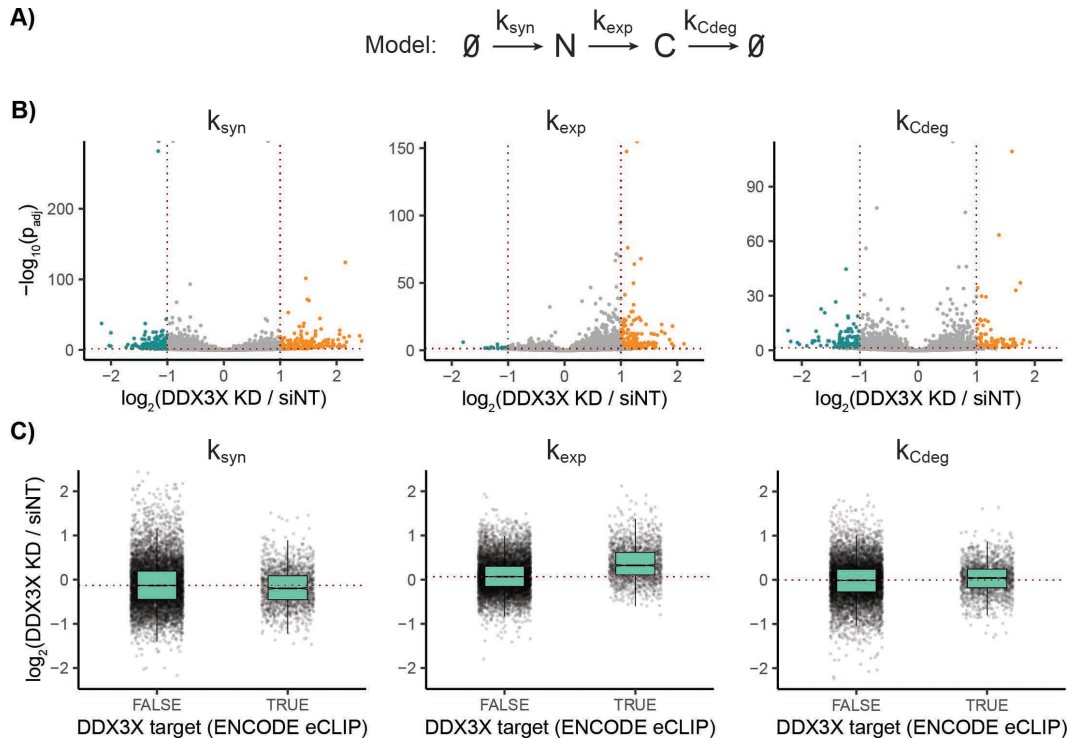

**Fig 7. EZbakR identifies the effects of DDX3X knockdown on subcellular RNA kinetics. A)** Model to which nuclear, cytoplasmic, and total RNA NR-seq data from Ietswaart et al. was fit [12]. **B)** EZbakR comparative analyses of synthesis, export, and degradation kinetics with and without DDX3X knockdown. **C)** Changes in synthesis, export, and degradation kinetics for DDX3X target and non-target transcripts. Red dotted lines represent median changes in each kinetic parameter for the non-targets.

probability for a user-specified range of L2FC values ([-0.25, 0.25] by default). The grandR (CI) strategy in Fig 6C-E calls significant genes for which the L2FC($k_{deg}$) 95% CI does not overlap 0. For the grandR (oracle) strategy, we identified a ROPE probability cutoff that yielded FDR control for each sample on-par with the best performing model (bakR MCMC) and used that cutoff to assess grandR's Power and MCC (cutoffs ranged from 0.83 to 0.87). The latter strategy is termed an "oracle" as it requires knowing the ground truth to finetune the cutoff. While this is not realistic in practice, it is a best-case scenario we used to assess the optimal efficacy of a ROPE-based decision strategy.

## INSPEcT analysis

For analyses in Fig C in S1 Text (panels A and B), INSPEcT was provided estimates of nascent and total RNA abundances, derived from either EZbakR's mixture model fit (panel A), or from calling reads with at least one mutation "nascent" (panel B). These read counts were then passed to INSPEcT's quantifyExpressionsFromTrCounts() function, whose output was passed to INSPEcT's newINSPEcT() and ratesFirstGuess() functions to obtain kinetic parameter estimates.

### Identifying DDX3X target transcripts

For analyses in Fig 7, DDX3X eCLIP data from ENCODE was used to identify targets of DDX3X [49]. Hits were identified as sites with at least an 8-fold enrichment in eCLIP read coverage over controls, and a p-value of less than 1e-10. The findOverlaps() function in the GenomicRanges package was used to identify genes harboring these hits (of which there were 1,673 unique genes), and this list of genes was cross-referenced with the relevant EZbakR analysis [50].

## Results

### Generalized feature assignment

In theory, reads in an NR-seq experiment can be assigned to any annotated genomic feature (e.g., genes), and the fraction of reads derived from labeled RNA can be estimated for each instance of that feature. In practice, existing pipelines provide limited flexibility in terms of the types of features to which they assign reads. To address this limitation, fastq2EZbakR implements a feature assignment strategy that is more flexible than any existing NR-seq analysis pipeline (Fig 2). In particular, reads can be assigned to genes (exons and introns), exclusively exonic regions of genes, exonic bins [51], transcript equivalence classes (TECs; i.e., the set of transcript isoforms with which a read is compatible) [52], and exon-exon junctions (Fig 2A). fastq2EZbakR also implements the read pre-processing, alignment, and mutation counting necessary for NR-seq data (Fig 2B). The EZbakR input data object is highly flexible (see EZbakR documentation and Fig 1, point 2). EZbakR accepts information about read assignment to any arbitrary combination of features and can perform analyses on any subset of these features (or the full set of features). This allows analyses performed by EZbakR to be easily integrated with that of expression analysis tools like DEseq2, edgeR, DEXSeq, etc., and opens the door for integration across different feature-level analyses (e.g., pre-mRNA and mature mRNA) [35,36,51]. The EZbakR suite thus generalizes the assignment of NR-seq reads to genomic features.

### Generalized and improved mixture modeling

After reads have been assigned to features and mutations in each read counted, the next step of any NR-seq analysis is to figure out how many sequencing reads are derived from labeled RNA for each feature (i.e., the fraction labeled, denoted $\theta$ in figures). While it is common to define a mutation content cutoff such that all reads with that many mutations or more are attributed to labeled RNA, this strategy can yield highly biased estimates (Fig A in S1 Text). Mixture modeling is a far more accurate, rigorous, and robust strategy, and has been implemented in several existing tools [3,31,34]. In this strategy, assumptions are made regarding distributions that accurately describe the probabilities of seeing N mutations in reads from labeled and unlabeled RNAs with M mutable reference nucleotides. Typically, this means assuming that a read's mutational content is binomially distributed, conditional on its status as being from labeled or unlabeled RNA. GRAND-SLAM and bakR both use this strategy to analyze single-label NR-seq data. In EZbakR, we have generalized this model to include analyses of multi-label NR-seq datasets (Fig 3A). To showcase this generalized model, we considered one such method, TILAC, where $s^4U$ labeled cells are mixed with $s^6G$ labeled cells (Fig 3B) [27]. TILAC allows for internally normalized comparative analyses, analogous to the proteomic method SILAC [53]. In TILAC, there are three populations of reads: those with high T-to-C mutation rates, those with high G-to-A mutation rates, and those with background levels of T-to-C and G-to-A mutation rates (Fig 3C). Reads with both high T-to-C and G-to-A mutation rates cannot exist in this case, as no cell was exposed to both labels. EZbakR is flexible and able to account for the absence of this fourth population (Fig 3C mutational populations table). Analyses of simulated TILAC data confirmed that EZbakR's multi-label mixture model fit is accurate (Fig 3D). The EZbakR suite thus generalizes mixture modeling of NR-seq data.

GRAND-SLAM and bakR make several additional assumptions regarding the mutational content of sequencing reads. Most significantly, they both assume that all RNA synthesized in the presence of metabolic label had an equal probability of incorporating said label. This is codified in the assumption that the distribution of mutation counts in reads from labeled

RNA is well described as a binomial distribution with one probability of mutation parameter ($p_{labeled}$) for all reads in a given sample. Recent work has shown that this assumption is violated by some transcript types. For example, mitochondrial RNAs have much lower labeled read mutation rates than other RNAs [32]. Because of this, we developed a strategy to allow each analyzed feature (e.g., each gene) to have its own incorporation rate parameter. To make this strategy robust, we employed a hierarchical modeling approach, in which we first infer a conservative feature-specific $p_{labeled}$ prior distribution and then use this to regularize the feature-specific estimates (Fig 4A and Design and Implementation). This prevents extreme estimate variance due to low feature coverage limiting the precision of the feature-specific incorporation rate estimation. Analyses of simulated data confirmed that this strategy provides accurate estimates of the fraction labeled and feature-specific $p_{labeled}$ (Fig 4B). Analyzing a real TimeLapse-seq dataset (reads assigned to exonic regions of genes; separate $p_{labeled}$ estimated for each gene) revealed that this strategy is able to identify and account for the previously-reported low incorporation rate for transcripts encoded on the mitochondrial chromosome (Fig 4C) [12,32]. While the hierarchical and non-hierarchical models provide similar fraction labeled estimates in most cases, the hierarchical model outputs far higher fraction labeled estimates for these mitochondrially encoded transcripts (Fig B in S1 Text). This is due to a sample-wide $p_{labeled}$ being an overestimate for these transcripts, effectively leading to the misclassification of many reads with moderate T-to-C mutation counts as unlabeled when using the non-hierarchical model (Fig 4C). The EZbakR suite thus improves mixture modeling of NR-seq data.

## Generalized linear dynamical systems modeling

Once the labeled read abundance for a given feature has been rigorously estimated, the next step of many NR-seq analyses is the inference of kinetic parameters. In a standard whole-cell NR-seq kinetic analysis, this means using the exonic fraction labeled and normalized read count to infer synthesis and degradation rate constants. Implicitly, this means assuming a model of the RNA dynamical system where mature mRNA is synthesized at a rate $k_{syn}$ and degraded with a rate constant $k_{deg}$ (Fig 5A). Recently, several studies have combined subcellular fractionation with NR-seq to investigate the rate of flow between subcellular compartments, as well as to probe the dynamics of RNA in each compartment [12–14]. In this case, the data needs to be fit to a more complicated model. To date, this has required in-house, specialized solutions for a given fractionation scheme and assumed dynamical system. A strategy to generalize this approach to support any experimental design and model of RNA population dynamics would greatly facilitate analyses of these data.

   Towards that end, EZbakR allows users to specify and estimate parameters of any identifiable, linear dynamical systems model of their choice. All such models, depicted as directed graphs in Fig 5, can be described using a system of ordinary differential equations (ODEs) with analytic solutions (see Design and Implementation and S1 Text for details). EZbakR infers the analytic solution for the dynamics of all modeled RNA species and uses this to perform maximum likelihood estimation of the kinetic parameters in the specified model. By combining generalized feature assignment in fastq2EZbakR with generalized linear dynamical systems modeling, EZbakR can model premature mRNA processing dynamics. Analyses of simulated data confirm that this parameter estimation procedure is accurate (Fig 5B). The accuracy of EZbakR is comparable to that of INSPEcT, a tool specialized for fitting this specific model (Fig C, panel A, in S1 Text) [54]. In addition, EZbakR significantly outperforms INSPEcT when INSPEcT is provided estimates of new and old RNA abundances from a naïve mutation cutoff strategy, rather than from the rigorous mixture modeling implemented in EZbakR (Fig C, panel B, in S1 Text). To test if EZbakR provides biologically reasonable kinetic parameter estimates when modeling pre-mRNA dynamics, we analyzed real NR-seq data from total RNA [12]. Despite the low absolute abundance of pre-mRNA, introns are typically much longer than exons, meaning that a significant portion of mapped reads in standard total RNA libraries originate from pre-mRNA (Fig C in S1 Text). EZbakR thus allows users to make full use of all these reads. The EZbakR analysis of this data revealed that pre-mRNA processing kinetics are typically much faster than mature mRNA degradation kinetics, which is a conclusion well corroborated by extensive previous work (Fig C panel C in S1 Text) [54–57].

EZbakR also supports analyses of RNA flow between subcellular compartments (Fig 5C, top) from experiments such as those performed by Ietswaart et al [12–14]. This requires integrating information across multiple independent samples from distinct RNA populations. Thus, using the read counts from these samples to inform kinetic parameter estimates would require a normalization strategy accounting for the differences in absolute molecular abundances across samples. While this is generally difficult, we drew inspiration from recent work and developed a normalization strategy that, with a sufficient set of samples, permits rigorous normalization [14]. This strategy relies on having samples from each individual fraction, as well as a necessary set of combinations of fractions (e.g., nuclear, cytoplasmic, and whole cell RNA). Scale factors can be estimated by modeling the total fraction of reads that are labeled in the combination of fractions (e.g., whole cell data) as a weighted sum of the same quantity in each individual fraction (e.g., nuclear and cytoplasmic fractions; see Design and Implementation for details). Analyses of simulated nuclear and cytoplasmic fractionation data confirm that this strategy is accurate (Fig 5C, bottom). In addition, modeling properly normalized read counts improves the accuracy of kinetic parameter estimates in simple models, while also facilitating investigation of more complicated models (Figs D and E in S1 Text). One challenge of fitting more complex models is that while all parameters may be theoretically identifiable, they may not be practically identifiable, e.g., due to the limited impact of a given parameter's precise value on measured quantities (Fig 5D; also see S1 Text for further discussion). EZbakR thus provides uncertainty quantification that can flag instances of practical unidentifiability, allowing users to avoid overinterpreting low confidence estimates (Fig 5D). Applying this strategy to a recent subcellular fractionation NR-seq dataset allowed us to corroborate previously estimated subcellular kinetic parameters while flagging the challenges of estimating polysome loading kinetics with this data (Fig F in S1 Text). The uncertainty quantification provided by EZbakR also facilitated robustly exploring the prevalence of nuclear degradation in a human cell line (Fig G in S1 Text). EZbakR corroborated many suspected genes producing a significant number of RNAs degraded in the nucleus (recently termed predicted to undergo nuclear degradation, or PUNDs), while also expanding the list of putative PUNDs (S1 Table) [12]. The EZbakR suite thus generalizes the kinetic modeling of NR-seq data.

## Generalized comparative analyses

Often, important biological conclusions are drawn not from kinetic parameter estimates in a given experimental condition, but from comparative analyses of these estimates across different biological conditions. bakR introduced a novel hierarchical model that significantly increased the power of comparative analyses of synthesis and degradation kinetics [34]. We have implemented this model in EZbakR, and it is compatible with all the kinetic parameter inference strategies in EZbakR. While bakR is only able to perform comparisons of multiple experimental conditions to a single reference condition, EZbakR implements a design matrix-specified generalized linear model that greatly increases the flexibility of its comparative analyses (Figs 6A and H, panel A, in S1 Text). In addition to providing greater model flexibility than its predecessor, EZbakR also improves upon the computational and statistical performance of both bakR and the GRAND-SLAM helper package grandR (Rummel et al. 2023). bakR includes three implementations of the same model, which differ in statistical rigor and computational intensiveness [34]. It includes a fast but conservative implementation (termed the MLE implementation) as well as more highly powered implementations that suffer from much longer runtimes (e.g., the MCMC implementation). In EZbakR, we have improved upon the efficient implementation in bakR to achieve statistical performance on-par with the more computationally intensive strategies in bakR, while not sacrificing efficiency (Fig 6B-E). In addition, EZbakR is orders of magnitude more efficient than grandR for this task (Fig 6B), while also having much higher statistical power (Fig 6C-E). Finally, EZbakR facilitates analyses of complex experimental designs using generalized linear modeling. For example, with a single model fit, EZbakR is able to perform any possible pairwise comparison in a recently published multi-perturbation NR-seq dataset (Fig 6F) [48]. In addition, EZbakR enables investigation of interaction effects (e.g., when the magnitude of a kinetic parameter's change upon one perturbation is dependent on the presence or absence of another perturbation), allowing it to quantify the extent to which the previously identified hypoxia driven RNA

destabilization is dependent on macrophage activation (Fig H, panel B, in S1 Text). The EZbakR suite thus generalizes performing well-powered and flexible comparisons of RNA kinetic parameters with NR-seq data.

EZbakR is designed to be able to dissect the impacts of biological perturbations on subcellular RNA kinetics by combining its generalized linear dynamical systems modeling with its generalized linear modeling. To test this, we analyzed published subcellular NR-seq data collected with and without siRNA knockdown of DDX3X (Fig 7A and 7B) [12]. EZbakR's comparative analyses of the estimated synthesis, export, and degradation kinetics corroborated the previously reported increase in export kinetics upon DDX3X knockdown (Fig 7B) [12]. Integrating these analyses with DDX3X eCLIP data from ENCODE revealed that the average increase in export kinetics is far higher for DDX3X targets than non-targets (Fig 7C) [49]. This is not the case for either the synthesis or degradation kinetics, suggesting that the export defects are the primary direct impact of DDX3X knockdown. These analyses confirm that the EZbakR suite provides high resolution insights about the subcellular kinetic impacts of biological perturbations.

## Discussion

To generalize and improve multiple aspects of NR-seq analyses, we have developed the EZbakR suite, which includes a Snakemake pipeline (fastq2EZbakR) and an R package (EZbakR) (Fig 1). Unlike existing tools, the EZbakR suite facilitates analyses of a wide array of genomic features (Fig 2), implements strategies for multi-label analyses (Fig 3), offers improved single-label modeling (Fig 4), can fit any identifiable linear dynamical systems model to NR-seq data (Fig 5), and implements an efficient and flexible generalized linear model-based kinetic parameter comparison strategy (Fig 6). The EZbakR suite thus promises to greatly facilitate deriving novel insights from NR-seq data.

The EZbakR suite will be useful for performing in-depth analyses of new NR-seq datasets and has the potential to uncover biological signal initially missed in existing NR-seq datasets. The limitations of current NR-seq bioinformatic tools have forced most analyses of total RNA NR-seq datasets to focus solely on the dynamics of mature mRNA. The EZbakR suite's generalized feature assignment strategy combined with its generalized linear dynamical systems modeling opens the door for investigating the kinetics of RNA processing in unprecedented detail. In addition, by performing analyses at the levels of splice junctions, exonic bins, or transcript equivalence classes, the EZbakR suite is able to identify isoform-level effects missed by standard gene-wide aggregate analyses [58]. In doing so, the EZbakR suite will allow researchers to make full use of their NR-seq data.

During the preparation of this manuscript, Halfpipe, a new NR-seq pipeline, was published [59]. While similar in many ways to SLAMDUNK, Halfpipe incorporates several notable enhancements. It provides access to the processed mutational data, implements a rigorous mixture modeling approach, and includes a strategy to fit a model of nuclear and cytoplasmic RNA dynamics. Despite this, Halfpipe has significant limitations compared to the EZbakR suite. It is specifically optimized for 3′-end sequencing data, does not implement a robust subcellular compartment normalization strategy, cannot fit alternative linear dynamical systems models, and does not support rigorous comparative analyses of kinetic parameter estimates across multiple biological conditions.

The EZbakR suite is designed to flexibly support analyses of the full universe of existing and future NR-seq extensions. From dynamical systems modeling of flow between subcellular compartments to analyses of multi-label NR-seq datasets, the EZbakR suite directly facilitates extension-specific analyses that no other tool supports. It is difficult for a single tool to perform all possible downstream analyses. For example, RNA velocity analyses of single cell NR-seq datasets both remain beyond the scope of the EZbakR suite and are supported by a number of existing specialized tools [20,23,24]. That said, all NR-seq analyses must start with the quantification of labeled RNA species. By improving models to perform these analyses and providing convenient access to the data required to fit such models, the EZbakR suite promises to improve analyses of all NR-seq datasets, regardless of their idiosyncrasies (e.g., unique data preprocessing requirements). In addition, the modular design of the EZbakR suite allows it to make use of the output of specialized pipelines for other metabolic labeling experiments (e.g., long read direct RNA sequencing methods) and will facilitate future

development and extension of its functionalities (Fig I in S1 Text) [47,60]. We anticipate that the EZbakR suite will provide a unified bioinformatic platform to support the ever-expanding NR-seq ecosystem.

**Availability and future directions**

fastq2EZbakR and EZbakR are both freely available on Github:

• fastq2EZbakR: https://github.com/isaacvock/fastq2EZbakR

• EZbakR: https://github.com/isaacvock/EZbakR

Scripts to reproduce all figures and supplemental figures are available on Github, at: https://github.com/isaacvock/EZbakRsuite_paper_code

Processed data necessary to reproduce all figures is available on Zenodo (https://doi.org/10.5281/zenodo.13929898), as well as supplemental tables of:

• All kinetic parameter estimates for analysis of total-cytoplasmic-nuclear NR-seq data (Fig G in S1 Text).

• All kinetic parameter estimates for analysis of premature mRNA and mature mRNA dynamics (Fig C in S1 Text).

Real NR-seq data analyzed in this paper were obtained from GEO (GSE207924 and GSE245750).

Future work on the EZbakR suite will include implementing a number of additional downstream analysis strategies specific to certain NR-seq extensions.

## Supporting information

**S1 Text.  Supplemental Figures and Methods.**
(DOCX)

**S1 Table.  Analysis of nuclear degradation in K562 cells (data from Ietswaart et al.).**
(CSV)

## Acknowledgments

The authors thank members of the Simon (namely Andreas Pintado-Urbanc and Michelle Moon), Pai (namely Jesse Lehman and Athma Pai), and Churchman (namely Robert Iestwaart, Erik McShane, Mary Couvillion, and Stirling Churchman) labs for useful discussions during the building and testing of EZbakR. We also thank the Churchman lab for early access to their subcellular TimeLapse-seq dataset. Finally, we want to thank all members of the Simon lab for helpful comments on this manuscript. This work was supported by the Intramural Research Program, National Heart, Lung, and Blood Institute, National Institutes of Health, and utilized the computational resources of the NIH HPC Biowulf cluster (http://hpc.nih.gov).

## Author contributions

**Conceptualization:** Isaac W. Vock, Martin Machyna.

**Data curation:** Isaac W. Vock.

**Formal analysis:** Isaac W. Vock.

**Funding acquisition:** J. Robert Hogg, Matthew D. Simon.

**Investigation:** Isaac W. Vock.

**Methodology:** Isaac W. Vock.

**Project administration:** J. Robert Hogg, Matthew D. Simon.

**Software:** Isaac W. Vock.

**Supervision:** J. Robert Hogg, Matthew D. Simon.

**Validation:** Isaac W. Vock, Justin W. Mabin, Alexandra Zhang.

**Visualization:** Isaac W. Vock.

**Writing – original draft:** Isaac W. Vock.

**Writing – review & editing:** Isaac W. Vock, Justin W. Mabin, Martin Machyna, Alexandra Zhang, J. Robert Hogg, Matthew D. Simon.

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
