## [Decision Letter · Decision Letter 0]

5 Feb 2025

PCOMPBIOL-D-24-01924

Expanding and improving analyses of nucleotide recoding RNA-seq experiments with the EZbakR suite

PLOS Computational Biology

Dear Dr. Vock,

Thank you for submitting your manuscript to PLOS Computational Biology. Your manuscript describing EZbakR has been reviewed by two experts in the field. While they acknowledge the potential value of your software package and its well-maintained codebase, both reviewers have identified several important issues that need to be addressed before the manuscript can be considered for publication. We therefore invite you to submit a major revision addressing the following key points: validation on real data, expanded comparative analysis, and improvements to presentation including terminology.<o:p></o:p>

Please submit your revised manuscript within 60 days Apr 07 2025 11:59PM. If you will need more time than this to complete your revisions, please reply to this message or contact the journal office at ploscompbiol@plos.org. Please include the following items when submitting your revised manuscript:

We look forward to receiving your revised manuscript.

Kind regards,

Quentin Gouil

Guest Editor

PLOS Computational Biology

Ilya Ioshikhes

Section Editor

PLOS Computational Biology

**Journal Requirements:**

3) Your manuscript is missing the following sections: Design and Implementation, and Availability and Future Directions. Please ensure that your article adheres to the standard Software article layout and order of Abstract, Introduction, Design and Implementation, Results, and Availability and Future Directions. For details on what each section should contain, see our Software article guidelines:

https://journals.plos.org/ploscompbiol/s/submission-guidelines#loc-software-submissions

5) We have noticed that you have uploaded Supporting Information files, but you have not included a list of legends. Please add a full list of legends for your Supporting Information files after the references list.

6) Please ensure that the funders and grant numbers match between the Financial Disclosure field and the Funding Information tab in your submission form. Note that the funders must be provided in the same order in both places as well.

**Reviewers' comments:**

Reviewer's Responses to Questions

**Comments to the Authors:**

Reviewer #1: In this manuscript, the authors present a new version of their software bakR, called EZbakR, for performing differential kinetics analysis on RNA metabolism.

I have no significant concerns regarding the methods or the implementation, the online documentation and the supplementary methods are relatively complete.

However, I believe that the manuscript itself is premature and lacks in scholarly presentation, and most results presented are simulated,

except for Figures 4C and related supplementary figures, unless I'm mistaken, or consist in schematics, without context or adequate description,

sometimes requiring an effort to understand how the figures relate to the text.

In particular, I have a few specific comments:

- In "generalized and improved mixture modeling", an example of feature-specific incorporation rate is shown using MT, but what other "features"

can be used, if any use-cases is known, and how would this approach generalize?

- For "generalized linear dynamical systems modeling" and "generalized comparative analyses", no real data are shown.

It is difficult to evaluate the capabilities of EZbakR. Uses cases of modeling flow between subcellular compartments is particularly interesting, but it's not clear how much of

data from 14 was used for this. I don't see any significant results for "generalized comparative analyses", and nothing regarding experimental perturbations, etc.

For this particular aspect, and in general for visualization, I don't see how EZbakR is better than grandR.

- In fact for most results presented in "generalized linear dynamical systems modeling", "fraction labeled mixing model" was used, this is the

ideal case if no spike-ins are available. But anyhow a comparison should be made with "fraction labeled modeling" and, ideally, with spike-ins.

- Overall, I believe well-designed uses-cases should be presented to validate the new features proposed in this work, either by using published data

with sufficient detail and relevance, either by producing new data, or at least for simulated data, by providing a comparative analysis with established

tools that may in some cases provide similar analyses such as grandR, or e.g. for modeling of premature and mature RNA expression, INSPEcT, etc.

I also have a number of rather more pedantic comments, but I don't understand why the authors are trying to redefine an already established terminology.

The authors coined the term "nucleotide recoding RNA sequencing methods or NR-seq" in a number of their own publications, e.g. 10.1093/nar/gkac693, 10.1261/rna.079451.122 or preprints such as 10.1101/2023.05.24.542133, etc. however this terminology leads to confusion, it is not generally used, and could be falsely understood to refer to the (biological) concept of recoding. The authors themselves describe NR-seq as experiments involving "chemically converting s4U from a uridine analog to a cytidine analog". So why not stick to common usage e.g. "nucleotide conversion", "4sU-tagging", or "s6G-tagging", etc. ? Besides, when they refer to NR-seq, in the context of this manuscript, i.e. for kinetics analysis/modelling, kinetic parameter estimation, they essentially refer to TimeLapse-seq, SLAM-seq, and TUC-seq, or their derivatives. These three methods are so far the most used metabolic labelling methods for these types of analysis. So why NR-seq? Why adding a "etc." to this list? Finally, a rather large paragraph of the introduction is dedicated to various methods (what they refer to as NR-seq) using s4U metabolic labeling in one way or another, but not necessarily within the context of kinetic modelling as referred to in this manuscript, e.g. references 8, 10, or 17 (authors cite their own work). Some references are even not relevant, i.e. not using 4sU-tagging at all, at least as far as I know, e.g. 11 and 13. I don't see why this is necessary, this just misleads the reader.

In the introduction, the authors mention pulseR and SLAMDUNK, and say that kinetic parameter estimation is "dependent on the metabolic label incorporation rate". They mention their own proposed method for TimeLapse-seq, which eventually is similar to the GRAND-SLAM approach. However, their

mixture model approach, that of pulseR, or GRAND-SLAM, all rely on some "mutational analysis", as they refer to, which ultimately boils down

to some form of identification of labeled vs. unlabeled. pulseR and GRAND-SLAM were previously compared across different labelling methods,

the authors should check this paper DOI: 10.1093/bib/bbab219.

The authors also say that "grandR [does not] support well-powered comparative analyses akin to those implemented in differential expression analysis software".

I must say that I have not used grandR, but looking at the documentation, I tend to disagree with the authors. grandR appears to be much

more flexible than GRAND-SLAM, allows to access data and metadata, including intermediate data, and perform various DE analyses. In all cases, the authors

fail to provide a fair comparison between EZbakR and grandR.

Reviewer #2: The authors developed an R package, EZbakR, and a corresponding Snakemake pipeline to expand the possible analyses for nucleotide recoding sequencing data such as NR-seq, TimeLapse-seq, SLAM-seq, TUC-seq, and others. The tool successfully estimates simulated parameters based on differential equations models derived from steady state kinetics. Following parameter estimation, the package can also perform generalized linear regression inference on any estimated parameter to identify possible changes between conditions. Overall, this software will be a valuable addition to facilitate the analysis of several existing and new datasets that aim to record the kinetics of RNA metabolism. Moreover, the authors have done an excellent job maintaining the codebase, ensuring it is well-structured and sufficiently documented for execution.

Major point:

1. The authors mostly use simulations to highlight accuracy. This is acceptable given the lack of a gold standard. They also provide some real data applications of the tool to showcase qualitatively meaningful results. Since the authors have used the data from Ietswaart et al., 2024, it is important to also provide comparisons to the methodology used in that publication. Currently only a small subset of this comparison is presented in Figure S6. This analysis needs to be expanded so the readers can evaluate expected differences between the two workflows.

Minor points:

2. Recently another tool, Nanodynamo, was published (PMID: 39231948, Nat. Comm) to perform similar analysis for long-read direct RNA sequencing with nanopores. While Nanodynamo has a different starting point (i.e. long reads), once the labeled and unlabeled reads are identified, one could use Nanodynamo to infer kinetic rates similar to EZbakR. Given that more tools are appearing to support NR-seq with direct RNA sequencing (PMID: 39211330) it would be beneficial for the authors to discuss EZbakR applicability to such datasets and potentially compare to Nanodynamo.

3. There are some minor grammatical issues throughout that need editing.

**Have the authors made all data and (if applicable) computational code underlying the findings in their manuscript fully available?**

Reviewer #1: Yes

Reviewer #2: Yes

PLOS authors have the option to publish the peer review history of their article (what does this mean? ). If published, this will include your full peer review and any attached files.

**Do you want your identity to be public for this peer review?** For information about this choice, including consent withdrawal, please see our Privacy Policy .

Reviewer #1: No

Reviewer #2: No

**Figure resubmission:**

**Reproducibility:**



---

## [Decision Letter · Decision Letter 1]

30 May 2025

Dear Mr. Vock,

We are pleased to inform you that your manuscript 'Expanding and improving analyses of nucleotide recoding RNA-seq experiments with the EZbakR suite' has been provisionally accepted for publication in PLOS Computational Biology.

Best regards,

Quentin Gouil

Guest Editor

PLOS Computational Biology

Ilya Ioshikhes

Section Editor

PLOS Computational Biology

Reviewer's Responses to Questions

**Comments to the Authors:**

Reviewer #1: In the carefully and thoroughly revised version of their manuscript, Vock et al. have addressed all of my issues and concerns.

Congratulations on such a nice piece of work

Reviewer #2: The authors have done an excellent job addressing the reviewers' concerns. I have no further comments. EZbakR will be a valuable tool for the community.

**Have the authors made all data and (if applicable) computational code underlying the findings in their manuscript fully available?**

Reviewer #1: Yes

Reviewer #2: Yes

PLOS authors have the option to publish the peer review history of their article (what does this mean? ). If published, this will include your full peer review and any attached files.

**Do you want your identity to be public for this peer review?** For information about this choice, including consent withdrawal, please see our Privacy Policy .

Reviewer #1: No

Reviewer #2: No

---

## [Editor Report · Acceptance letter]

PCOMPBIOL-D-24-01924R1

Expanding and improving analyses of nucleotide recoding RNA-seq experiments with the EZbakR suite

Dear Dr Vock,

I am pleased to inform you that your manuscript has been formally accepted for publication in PLOS Computational Biology. Your manuscript is now with our production department and you will be notified of the publication date in due course.

With kind regards,

Anita Estes
